# How to attract and retain health workers in rural areas of a fragile state: Findings from a labour market survey in Guinea

**Sophie Witter**[1,2]*, **Christopher H. Herbst**[3], **Marc Smitz**[2], **Mamadou Dioulde Balde**[4], **Ibrahim Magazi**[3], **Rashid U. Zaman**[2]

**1** Institute of Global Health and Development & ReBUILD for Resilience, Queen Margaret University, Edinburgh, United Kingdom, **2** Oxford Policy Management, Oxford, United Kingdom, **3** World Bank, Washington, DC, United States of America, **4** Cellule de Recherche en Santé de la Reproduction en Guinée (CERREGUI), Conakry, Guinea

* switter@qmu.ac.uk

**Data Availability Statement:** All relevant data are within the paper and its Supporting information files.

## Abstract

Most countries face challenges attracting and retaining health staff in remote areas but this is especially acute in fragile and shock-prone contexts, like Guinea, where imbalances in staffing are high and financial and governance arrangements to address rural shortfalls are weak. The objective of this study was to understand how health staff could be better motivated to work and remain in rural, under-served areas in Guinea. In order to inform the policy dialogue on strengthening human resources for health, we conducted three nationally representative cross-sectional surveys, adapted from tools used in other fragile contexts. This article focuses on the health worker survey. We found that the locational job preferences of health workers in Guinea are particularly influenced by opportunities for training, working conditions, and housing. Most staff are satisfied with their work and with supervision, however, financial aspects and working conditions are considered least satisfactory, and worrying findings include the high proportion of staff favouring emigration, their high tolerance of informal user payments, as well as their limited exposure to rural areas during training. Based on our findings, we highlight measures which could improve rural recruitment and retention in Guinea and similar settings. These include offering upgrading and specialization in return for rural service; providing greater exposure to rural areas during training; increasing recruitment from rural areas; experimenting with fixed term contracts in rural areas; and improving working conditions in rural posts. The development of incentive packages should be accompanied by action to tackle wider issues, such as reforms to training and staff management.

## Introduction

Health professionals are a fundamental component of health service delivery. Their numbers, distribution, and performance directly affect health outcomes. Global studies show statistically significant associations between the density of health workforce and maternal, infant, and under-five mortality rates, and vaccination coverage [1, 2].

**Funding:** This article reports on work funded by the World Bank. World Bank members contributed to study design but the content of the article are the responsibility of the authors alone.

**Competing interests:** This study was funded by a grant from the World Bank. The grant provided support in the form of salaries for the authors, including authors directly employed by the World Bank, and the larger study team, and also funded all field work related expenses. Aside from funding individuals involved in the study, the grant was also used to hire Oxford Policy Management, an international development consulting firm, to support overall study design, data collection and analysis. The specific roles of all authors are articulated in the 'author contributions' section Affiliation with the World Bank, Oxford Policy Management, or any other entity involved in the study, does not alter our adherence to PLOS ONE policies in sharing data and materials Please note that the findings, interpretations, and conclusions expressed in this work do not necessarily reflect the views of The World Bank, its Board of Executive Directors, or the governments they represent. The World Bank does not guarantee the accuracy, completeness, or currency of the data included in this work and does not assume responsibility for any errors, omissions, or discrepancies in the information, or liability with respect to the use of or failure to use the information, methods, processes, or conclusions set forth. The boundaries, colors, denominations, and other information shown on any map in this work do not imply any judgment on the part of The World Bank concerning the legal status of any territory or the endorsement or accep¬tance of such boundaries'.

In 2016 the Human Development Index ranked Guinea 183rd out of 188 countries [3]. Guinea has a low life expectancy of 58 years, driven by high mortality ratios. Maternal mortality is still amongst the highest in the region [4]. The Guinea health system is facing the challenges of high need, low funding, and maldistribution of resources (including staff), which are clustered in urban areas. Public expenditure on health increased post-Ebola but remains very low, at an estimated US$7.58 per capita [5].

Guinea's health sector is organised into three levels [6]. The primary level is composed of 925 health posts and 410 health centres, which are the closest facilities to communities, and are predominantly found in rural areas. The secondary level is composed of 38 regional and prefectural hospitals that are respectively the first- and second-level referral hospitals for health centres. These tend to be located in regional and prefectural urban centres. The tertiary level has three specialised tertiary-level hospitals in Conakry. These hospitals cater largely to urban populations and the more economically advantaged. The parastatal health sector comprises three dispensaries and four hospitals associated with mining and agricultural enterprises. The private health sector consists both of non-profit and for-profit health institutions. It is mostly run by nurses and technical health agents (ATS—an auxiliary medical group trained to diploma level for two years), offering basic health services and treatment at varying prices. The formal private health sector consists of 41 clinics and 106 medical cabinets, as well as providers of traditional medicine.

The Government of Guinea has identified inequities in the composition and distribution of the health workforce, as well as the low motivation and sub-optimal performance of health professionals, as a critical challenge to the effective delivery of health services. At the end of 2014, there were 11,527 employees in public and private health facilities [6]. Of these, 7,706 can be classified as health workers, including doctors, nurses, midwives, and ATS. This represents a density of 0.62 health workers per 1,000 population, which indicates a large shortfall relative to the World Health Organisation minimum target of 2.28 per 1,000 population [7]. Urban areas are home to most of the hospitals, as well as private sector and additional income-generating opportunities (formal and informal). As a result, only 17% of the public sector health staff serve in rural areas, where 70% of the total population lives. Health centre provision is dominated by ATS. There are also concerns about quality of care provided in facilities, which often lack basic infrastructure, especially in rural areas [6].

The objective of this study was to understand how health staff could be better motivated to work and remain in rural, under-served areas in Guinea, in order to inform the policy dialogue (on strengthening human resources for health). This is a question which many countries struggle with [8–10], but which has particular urgency in fragile, shock-affected contexts like Guinea, where health needs are high, resources are limited, governance to ensure even and well-managed services across a large territory is constrained, and the divergence between living conditions in metropolitan areas and the rural hinterland are especially pronounced [8, 11].

## Materials and methods

### Approach and tools

We conducted three nationally representative cross-sectional surveys, adapted from tools used in other fragile contexts [12, 13]. For a summary of the findings from the facility survey and the exit interviews with patients, see the S1–S3 Files. In this article we focus on the health professionals survey. This engaged with general physicians, nurses/ midwives, and ATS employed in the primary-, secondary-, and tertiary-level health facilities in Guinea. It included generating information on the motivation, satisfaction, and preferences of health professionals

employed in the health labour market, with different backgrounds and characteristics. It included a Discrete Choice Experiment (DCE) to identify and measure the relative importance of variables influencing the uptake of different jobs, including in rural areas.

Structured questionnaires were used. Unprompted response was obtained where it was required (e.g. data for Figs 2 and 5). However, the response categories were read aloud where there were no additional biases of doing so (e.g. data for Figs 3 and 4).

## Discrete Choice Experiment (DCE)

A DCE consists in asking health workers to make multiple choices, each time between two (or more) jobs. Each job is valued according to different attributes, which vary in their levels. Assuming health workers reveal their true preferences, it enables the researcher to infer the relative importance of each attribute compared to the others.

The choice of job sets was informed by preliminary qualitative work conducted in January 2017, which included focus group discussions with medical doctors, medical students, nurses, midwives, and ATS in eight institutions (46 participants in total). These discussions focused on what attributes matter in relation to job choices, and what considerations were important for the health workers in relation to working in rural areas (Table 1).

Health professionals across four cadres were asked to choose between hypothetical jobs, according to their preferences. Since we had many job attributes, we chose to create 16 choices per health worker to be able to estimate the preferences completely (1st level interactions between attributes). The hypothetical set of jobs pairs was designed using a SAS macro % MktEx, which ensures that variance of the experimental matrix is maximised [14].

**Table 1. Selected Discrete Choice Experiment attributes.**

| Attribute | Levels and description |
|---|---|
| Location | Conakry hospital, regional hospital, or prefectural hospital. The latter was the lowest-level facility where doctors needed to be incentivised to be posted. For nurses and midwives, the lowest facility level was rural health centres (4 levels). For technical health agents (ATS), the health posts were the lowest level (5 levels). |
| Equipment | For the infrastructure and equipment, we included a superior option which included a waiting room, a private room for health workers (with a phone and a work computer), as well as equipment appropriate for different levels of facilities (e.g. echocardiography or ultrasound machine for national and regional hospitals). The intermediate option entailed sufficient smaller instruments and supplies that all facilities should have. These include thermometers, blood pressure meters (sphygmomanometer), stethoscopes, syringes, needles, stiches, bandages, and basic drugs. The lowest option consisted in the systemic lack of infrastructure and even small equipment. This is what is reported as the current field conditions in remote areas. |
| Housing | This was included as a dichotomous variable: whether housing was provided by the health facility administration or not. |
| Transport | We included a motorbike provided by the health facilities as the mode of transportation in the choice sets. This was a dichotomous variable. |
| Time-bound contract | This was the maximum duration the health workers would stay at the facility before being guaranteed a promotion or transfer to a higher-tier facility. For medical doctors, we selected five years, and for nurses, midwives and ATS we selected seven years. |
| Training | Medical doctors had choices of specialisation or attending workshops. Nurses and midwives had workshops on specific themes, or a technical specialisation. ATS were offered the opportunity to become a nurse through a specific programme. |
| Wage | Wage expectations for remote places were very high and had a large variability so it was decided to have two levels, with the second increment being larger than the first one. |

## Sampling

The sampling frame included health workers (doctors, nurses, midwives, and ATS) working in public sector health facilities in Guinea. We used a multi-stage stratified sampling approach to sample the prefectures and health facilities, and then to sample health workers and patients from the sampled health facilities. The calculated sample size for the health workers survey was 600 health workers and 480 patients. The requirement for a higher sample for the health workers was driven by the need to adjust for intra-cluster correlation related to the four different types of health workers. (The statistical assumptions used were: population size: 6,000; expected frequency: 50%; confidence level: 95%; margin of error: 5%; design effect: 1.0; number of clusters: 120; cluster size: 5).

In the first stage, we sampled 15 out of 33 prefectures in the country. The sampled prefectures include the capital Conakry, the prefecture containing the administrative centre of each of the seven regions, and another rural prefecture in each region. The rural prefectures were randomly sampled with equal probabilities from among all prefectures in the region.

In the second stage, we sampled health facilities including hospitals, health centres and health posts. A total of 55 health facilities were sampled, including three national hospitals, seven prefectural hospitals, seven regional hospitals, three communal hospitals, 28 health centres, and seven health posts. The sampled 17 hospitals included three national hospitals in Conakry, the seven regional hospitals in seven sampled regions, and the seven prefectural hospitals in the selected prefectures. We also sampled three community hospitals and health centres in Conakry. These were randomly selected with equal probability from the six such health facilities in the capital. In each of the other 14 selected prefectures, we sampled one urban health centre and one rural health centre, randomly selected with equal probability from among the total number or urban and rural health centres located in the prefecture. Among all 55 facilities, 10 were in the forested region, 17 in the maritime region, 17 in middle Guinea, and 11 in upper Guinea. In total, 34 were in urban areas and 21 in rural areas, reflecting the distribution of health infrastructure.

At the third stage we sampled the health workers and patients by sampling two or three *clusters* (each composed of five health workers), depending on the size of the facilities, in each of the 55 sampled facilities. In total, 120 clusters (600 health workers) were created within the 55 sampled facilities with higher number of clusters in larger facilities. Similarly, from each cluster we sampled four patients to reach the total sample of 480 patients. The field team members were given sampling forms in sealed envelopes containing the first random number and sampling steps so that they could draw the samples in the field.

## Fieldwork

The survey tools were administered using computer-assisted personal interviews (CAPI) in Android tablets. The fieldworkers' manual and the data collection tools were translated into French.

Enumerators received a two-week training on administering the data collection tools. They also carried out a pilot test at the end of the training. Data were collected over two months from December 2017 to January 2018. Data were uploaded every day to a portal, where it was checked for completeness and accuracy each day by a data management expert. Any discrepancies were resolved immediately while the teams were still in the field.

## Analysis

Data were transferred to Stata using StatTransfer. Data were then further checked, prior to the analysis, for ranges, outliers and internal inconsistency, and were cleaned.

We analysed data using R and Stata. Multiple regression and two sample tests for proportions were used to see statistical significance. We used multiple linear regression model to explain the variability due to covariates such as sex, age etc. We have created a composite score of satisfaction using 22 indicators, ranging from health workers' feeling of preparedness, workload, health facility management, and working conditions, and their finances. The score ranges from 1 to 20, with higher values representing greater dissatisfaction. The DCE regression was performed by fitting a conditional logit model. Statistical tests were considered significant at 95% confidence level with a p-value of less than 0.05.

Descriptive statistics were presented for health workers, patients, and health facility data by different attributes, including geographical regions, locality (urban–rural), health facility types, and health worker types.

During analysis, we grouped the health workers into 'highly motivated' and 'less motivated' categories, as well as 'mainly intrinsically motivated' and 'mainly extrinsically motivated' categories, based on a composite score for their motivation profile, consisting of 21 explicit indicators (for full details, see [15]). Those who responded more strongly to questions about the inherent value of their work and their motivation to serve the community were judged to be more intrinsically motivated.

The grouping of health workers according to their levels of satisfaction was done based on a composite score, consisting of measures of a) their confidence and reported efficacy, b) workload, c) work environment, c) financial situation, and e) four generic satisfaction scales concerning their career.

We used Guinean Franc (GNF) as the currency for the DCE tool and presented the findings in United States Dollar (USD) (at the exchange rate from August 2018 of 1 USD = 9,014 GNF).

## Ethics

The study protocols were approved by Oxford Policy Management prior to any data gathering and by the National Ethics Committee for Health Research of the Ministry of Health in Guinea before fieldwork was undertaken in October 2017. Informed consent was obtained in writing from all respondents prior to data collection by the data collectors and data were anonymised prior to analysis.

## Results

In the health worker survey, we sampled 600 health workers. One of the interviews was incomplete, leaving a sample of 599 health workers, including 153 (26%) doctors, 111 nurses (19%), 74 midwives (12%), and 261 (44%) ATS. Out of 153 doctors, 22 (4% of the entire sample) were specialist doctors. 57% were female and 43% male. Some of the key characteristics of the sampled health workers, including previous workplaces, mean years of working, workload and satisfactions scores in a four-point scale are included in Table 2.

### Health worker location and career history

Only 16% of health workers worked in rural areas, reflecting the national maldistribution of staffing. 95% were permanent employees.

Among the doctors, 96% were trained in Conakry and only 28% practised in rural areas during their training (Fig 1). Those working in rural areas were significantly more likely to have had exposure to rural areas during training (49% versus 39%). Nurses and midwives trained in two to three sites, and 44% and 35%, respectively, practised in rural areas during

**Table 2. Key characteristics of the health workers, by type.**

|  | Doctor | Nurse | Midwife | ATS |
|---|---|---|---|---|
| **Experience of working in various levels of health facilities** | | | | |
| National hospitals | 69.9% | 42.3% | 33.8% | 14.2% |
| Regional hospitals | 62.7% | 45.0% | 41.9% | 43.3% |
| Prefectural hospitals | 52.3% | 42.3% | 51.4% | 46.4% |
| Health centres | 22.9% | 65.8% | 58.1% | 70.9% |
| Health posts | 2.0% | 18.9% | 8.1% | 31.0% |
| Private health facilities | 26.1% | 22.5% | 29.7% | 11.9% |
| **Mean years of working** | | | | |
| As a health professional | 15.0 | 16.2 | 8.4 | 15.4 |
| At the current facility | 7.8 | 8.2 | 3.7 | 8.2 |
| **Mean statistics on workload** | | | | |
| Working hours per week | 53.4 | 50.7 | 47.9 | 50.7 |
| Patients per day | 8.3 | 7.8 | 7.2 | 7.8 |
| **Mean satisfaction scores by key areas (in a 4-point scale)** | | | | |
| Life as a whole | 2.7 | 2.7 | 2.7 | 2.8 |
| Career prospect | 2.5 | 2.6 | 2.7 | 2.6 |
| Work-life balance | 2.6 | 2.5 | 2.6 | 2.6 |
| Working conditions | 2.2 | 2.4 | 2.3 | 2.5 |
| Financial | 2.0 | 2.0 | 2.0 | 2.1 |
| **N** | **153** | **111** | **74** | **261** |

their training. For ATS, although their training occurs in the widest range of sites, just under half (47%) practised in rural areas during training.

This was also reflected in their work practice—only 9% of the sampled doctors had worked in rural areas for more than 10 years, while this proportion was even lower for midwives (6.5%). 25% of nurses and 28% of ATS had worked in rural areas for ≥10 years. Current work location reflected work experience: 86% of staff who worked in rural areas had previously worked in rural areas, compared to only 38% for those who worked in urban areas. Staff in urban facilities were significantly more experienced and remained in their posts longer (mean: 8.2 years in workplace at the time of the study) than staff in rural areas (only 3.2 years).

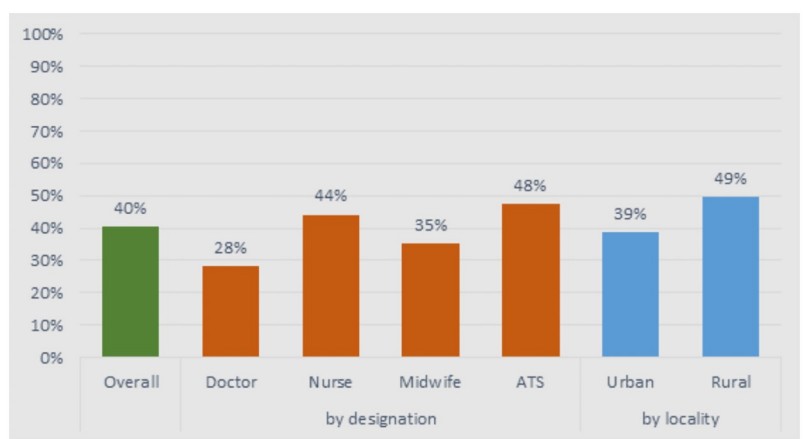

**Fig 1. Proportion of health workers who practised in rural areas during training, by health workers type.**

**Motivation to join the profession and preparedness.** "Helping others" was the overwhelming majority of the reported reasons for becoming health workers for all four cadres, with no significant difference between those in rural and urban areas (Fig 2).

Typically, health workers felt well prepared for their job (78%–88% across groups). However, 12% of doctors felt not at all prepared. 49% of health workers in rural areas indicated that they had to perform tasks "daily" or "often" for which they felt inadequately prepared, compared to 39% for urban staff (Fig 3).

**Workload.** Most staff (61–89% across the groups) felt that their colleagues were working as hard as they were, and there were no noteworthy differences in this across rural and urban areas. This was reflected in reported working days and hours, which were relatively uniform, at around 8.5 hours per day and six days per week across the cadres. Rural health staff worked significantly longer hours per week (58 versus 50 for urban health staff) and saw a higher number of patients per day but the difference was not significant (8.4 versus 7.7 for urban health staff). Only a small proportion of staff (6% in rural areas and 3% in urban areas) felt that they had a workload that was more than they could handle (Fig 4).

**Satisfaction.** The average health worker was quite satisfied with life in general (a score of 2.8, out of 4), and felt neutral satisfaction about their career prospects (2.6) and work–life balance (2.6). Health workers were the most unsatisfied with their financial situation (2.1) and

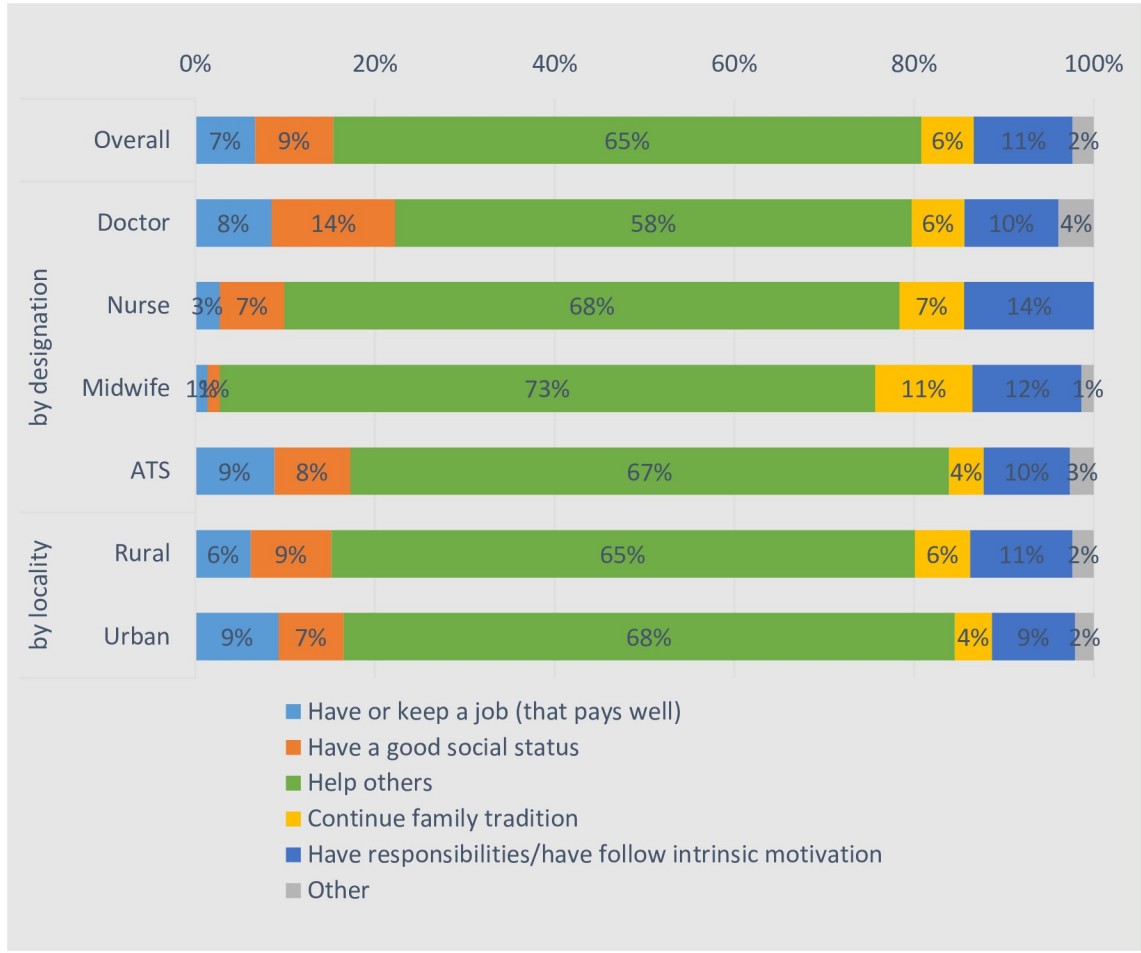

**Fig 2. What was the main reason for you to become a health worker?**

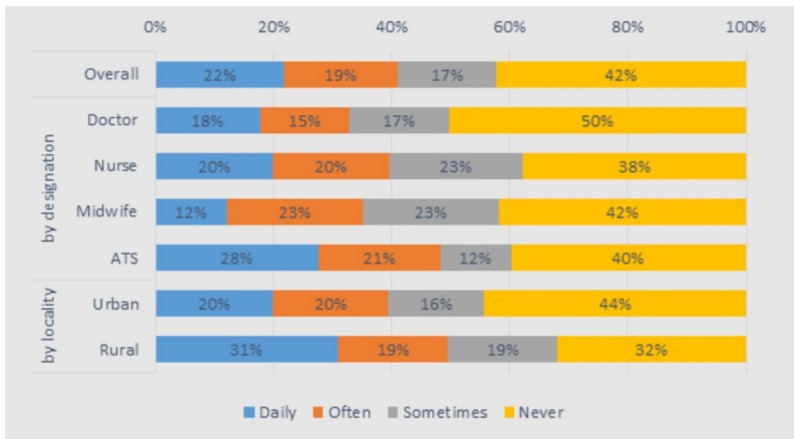

**Fig 3. Are you sometimes performing tasks which you were not prepared for in health professional training?**

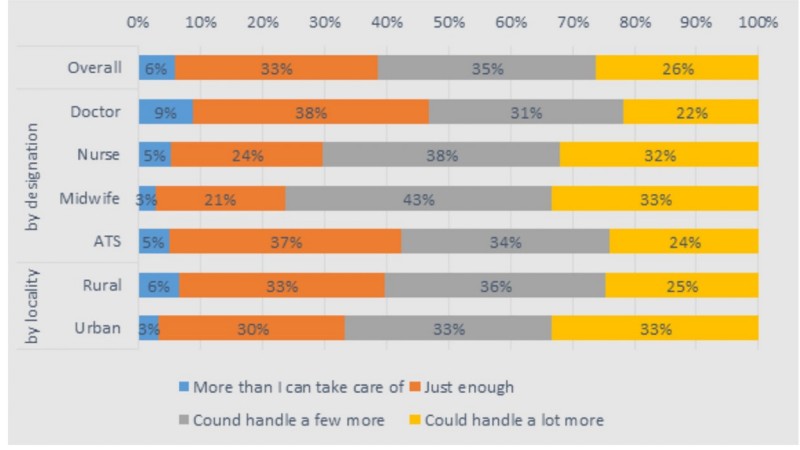

**Fig 4. How do you feel about the number of patients you are seeing in a day?**

then their working conditions (2.4), although this was mainly felt by doctors and midwives. There was no statistically significant difference between urban and rural workers in any of these areas.

A composite index covering 22 indicators suggests greater dissatisfaction for doctors, compared to other cadres, and a significantly higher level of dissatisfaction for rural staff (11.5 versus 10.3 for urban staff, out of 20). Predictors of reported general satisfaction with life included good working conditions, being intrinsically motivated, and coming from a better socio-economic background. In relation to the composite satisfaction score, working in urban areas and being female were also significantly and positively associated with satisfaction (Table 3).

On specific questions about their career, most of the mean scores indicate high satisfaction or positive feelings. The exceptions were worries about being unemployed or being posted elsewhere, and agreement with the statement that it is difficult for health workers to care much about whether the work that they do was done correctly or not.

**Table 3. Multiple regression model for predicting satisfaction composite score and overall satisfaction with life.**

| | Stated dissatisfaction with life overall | | | Dissatisfaction composite score | | |
|---|---|---|---|---|---|---|
| | est | SE | p-value | est | SE | p-value |
| is female | -0.007 | 0.055 | 0.897 | -1.38 | 0.39 | 0.010*** |
| age | 0.002 | 0.023 | 0.931 | 0.13 | 0.09 | 0.174 |
| age squared | 0.000 | 0.000 | 0.721 | 0.00 | 0.00 | 0.088* |
| has a spouse or is engaged | 0.022 | 0.085 | 0.803 | 0.18 | 0.43 | 0.693 |
| number of children | -0.022 | 0.013 | 0.131 | 0.02 | 0.06 | 0.806 |
| family background (socio-economic quintile) | 0.065 | 0.027 | 0.049** | -0.13 | 0.20 | 0.540 |
| currently works in rural facility | -0.062 | 0.097 | 0.544 | 1.02 | 0.41 | 0.0410** |
| currently works in hospital | -0.097 | 0.057 | 0.132 | -0.09 | 0.56 | 0.881 |
| is a nurse | -0.106 | 0.146 | 0.493 | -0.31 | 0.26 | 0.274 |
| is a midwife | 0.002 | 0.140 | 0.989 | -0.55 | 0.57 | 0.368 |
| is an ATS | 0.022 | 0.064 | 0.747 | 0.00 | 0.67 | 0.994 |
| income (US$) | -0.000 | 0.000 | 0.197 | 0.00 | 0.00 | 0.947 |
| income squared (US$) | 0.000 | 0.000 | 0.111 | 0.00 | 0.00 | 0.913 |
| number of years as a health worker | 0.000 | 0.002 | 0.852 | -0.01 | 0.01 | 0.652 |
| extrinsic motivation (external) | 0.048 | 0.035 | 0.215 | 0.22 | 0.19 | 0.298 |
| extrinsic motivation (introjected) | 0.003 | 0.045 | 0.943 | -0.07 | 0.13 | 0.622 |
| extrinsic motivation (integrated) | 0.064 | 0.035 | 0.109 | -0.34 | 0.22 | 0.170 |
| intrinsic motivation | 0.086 | 0.025 | 0.012** | -0.67 | 0.13 | 0.001*** |
| supervisor considers worker needs | 0.033 | 0.050 | 0.532 | -0.72 | 0.36 | 0.086* |
| facility equipment is good | 0.083 | 0.024 | 0.009** | -4.03 | 0.36 | <0.001*** |
| N | 572 | | | 572 | | |
| r2 | 0.1016 | | | 0.521 | | |

* $p<0.1$,

** $p<0.05$,

*** $p<0.01$.

The majority (60%) of health workers felt that the facility met the demands of their patients. However, doctors were the most critical, with only 37% feeling so. Health workers wanted to see improvements most in the skills of their colleagues (35%), the equipment (29%), and the infrastructure of their facilities (15%). While there was little variation among the four cadres, there were some significant differences between urban and rural health workers: 15% of rural workers wanted to see the availability of medicine and materials improve, compared to 6% of urban workers. Also, 9% of urban workers wanted to see the cleanliness and hygiene of their facilities improve, compared to 4% of rural health workers (Fig 5).

**Career plans and preferences.** A large majority (85%) of health workers preferred to work in urban areas in the longer term (Table 4). However, this average was much higher for doctors (94%), nurses (91%), and midwives (91%), while it was lower of ATS workers (75%). As expected, a greater proportion of those who worked in urban areas preferred to work in urban areas in the longer term (89%) than those who worked in rural areas (63%). Furthermore, a large proportion of health workers plan to migrate abroad (85%). This was consistent across the cadres and between urban and rural workers, with no significant difference between groups. 73% of health workers wished to transfer from their position in the near future, and, again, this was quite consistent across the cadres and localities. Of the 11% health workers that wanted to change professions, most said it was because they were looking for better career options or income prospects.

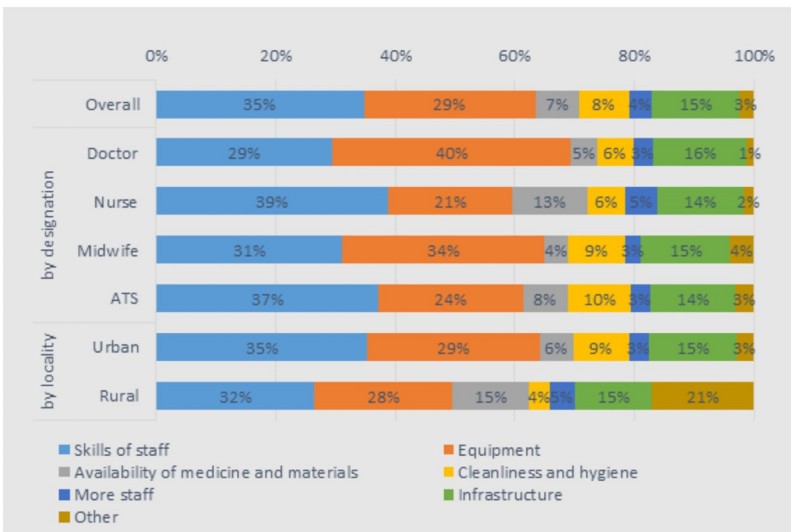

**Fig 5. If you could improve one of the following aspects in your health facility, which do you feel is the most important?**

**Table 4. Health worker preferences (location, position, and profession).**

|  | by designation |  |  |  |  | by locality |  |
|---|---|---|---|---|---|---|---|
|  | Overall | Doctor | Nurse | Midwife | ATS | Urban | Rural |
| **Prefers urban areas** | 85% | 94% | 91% | 91% | 75% | 89% | 63% |
| **Looking to migrate abroad** | 84% | 82% | 88% | 86% | 82% | 83% | 88% |
| **Prefers to transfer to new position** | 73% | 76% | 75% | 77% | 70% | 72% | 80% |
| **Looking to change profession** | 11% | 7% | 3% | 11% | 16% | 10% | 15% |

Those who intended to continue working in the health sector (89%) preferred to stay in the public/government sector (77%), while 10% preferred to switch to the private section and 14% to the non-government organisation/faith-based sector. A higher proportion of midwives preferred to stay in the public section (89%), compared to other workers, while doctors were more likely to want to switch to the private or non-government organisation /faith-based sector. There was little difference between those working in rural and urban areas in terms of preferring to stay in the public section (76% and 80%), while more urban workers preferred to switch to the private sector (11%) than rural (4%). Preferences by facility type largely reflect the distribution of staff across facility types.

If health care workers had prior experience of working in rural areas, they had a 14.6% greater likelihood of working there (Table 5). Each additional year of prior experience working in a rural facility was associated with a 1% increase in likelihood of preferring to work in rural areas in the long term. However, it was interesting to note that having some exposure to rural areas during training did not have a strong correlation with the fact that they were working in rural areas but had some correlation with a preference to do so. Another predictor of working in and preferring rural facilities was a health worker's motivation. Those with higher intrinsic motivation were more likely to be working in urban facilities (3% more likely)–perhaps because of the better working conditions found there—but were most likely to prefer rural facilities in the longer term.

**Table 5. Multiple regression model for predicting whether health worker currently works in and has long-term preference to work in rural health facility.**

| | Currently working in rural area | | | Has long-term preference to work in rural area | | |
|---|---|---|---|---|---|---|
| | est | SE | p-value | est | SE | p-value |
| is female | -0.029 | 0.028 | 0.330 | -0.016 | 0.016 | 0.340 |
| age | -0.026 | 0.011 | 0.043** | -0.028 | 0.016 | 0.130 |
| age squared | 0.000 | 0.000 | 0.147 | 0.000 | 0.000 | 0.215 |
| has a spouse or is engaged | 0.050 | 0.040 | 0.249 | 0.018 | 0.041 | 0.676 |
| number of children | -0.001 | 0.009 | 0.882 | 0.012 | 0.008 | 0.158 |
| socio-economic family background (1–5) | -0.027 | 0.013 | 0.083* | -0.021 | 0.012 | 0.131 |
| grew up in rural area | -0.001 | 0.052 | 0.984 | 0.046 | 0.038 | 0.271 |
| exposed to rural areas during training | -0.045 | 0.047 | 0.369 | 0.039 | 0.018 | 0.064* |
| has experience in rural area | 0.146 | 0.066 | 0.064* | 0.017 | 0.044 | 0.706 |
| number of years of experience in rural area | 0.024 | 0.004 | <0.001*** | 0.010 | 0.004 | 0.034** |
| extrinsic motivation (external regulation) | 0.030 | 0.017 | 0.134 | -0.029 | 0.014 | 0.071* |
| intrinsic motivation | -0.028 | 0.012 | 0.043** | 0.011 | 0.006 | 0.113 |
| currently works in rural area | | | | -0.101 | 0.042 | 0.045** |
| N | 592 | | | 592 | | |
| r2 | 0.267 | | | 0.130 | | |

Note: Robust standard errors clustered by region;

*p<0.1,

**p<0.05,

***p<0.01.

**Remuneration.** Health workers had a mean monthly net salary of 185 USD and median of 156 USD, ranging from 127 USD for rural ATS to 551 USD for rural doctors. On average, health workers received an additional public payment of 19 USD (median of 2 USD), although this low value was due to 32% of health workers not receiving any other kind of non-salary payment.

Doctors were more likely than other health workers to receive additional non-salary payments, with 68% of urban doctors and 60% rural doctors receiving other payment. In particular, they were more likely to receive transport, housing, and other allowances. Nurses, midwives, and ATS workers received similar treatment in terms of getting allowances and other non-salary payments. Overall, there did not appear to be a consistent trend as regards urban and rural health workers.

Looking at payment amounts for the various non-salary payments, there was very little variation across cadre and locality. The median monthly payment for transport was 30 USD and for housing it was 11 USD, and these values were the same for all eight cadre and locality combinations. Perdiem and performance-based pay had more variation, as would be expected. Altogether, the median total non-salary payment was 45 USD, with the medians similar across cadre and locality.

Health workers received payments regularly, with 95% never experiencing a missing payment. However, one-third have been paid late. The average number of late payments in the last year was 1.5. The estimate was higher for doctors (1.8) and midwives (2.0), and higher for rural workers (2.2) compared to urban (1.3).

16% of health workers claimed that it was common practice for patients to pay for their services. This proportion was higher in rural areas (22%) than urban areas (15%), although the difference was not significant. 53% of health workers were comfortable with accepting payment and this was higher for rural workers (62%) than urban workers (51%), though not

significantly so. A range of other remuneration strategies were also pursued. Overall, 14% of health workers said it was common practice to receive payments to receive better treatment in public facilities, with an average of 14 USD received by health workers from this source. The proportion was higher in rural facilities (22%) than urban facilities (13%). Receiving gifts was widely prevalent, with 60% of health workers receiving gifts, and this was higher in urban facilities (62%) than rural ones (49%). Of those who said it was common, 88% were comfortable with the practice.

In terms of pursuing other economic activities, 23% of health workers said it was common practice to privately consult patients outside of work hours, and to receive an average of 10 USD per week doing so. Unexpectedly, the proportion was higher among rural workers (36%) than urban workers (20%). 21% of health workers also say it was common practice to pursue non-health-related income-generating activities, like farming, in addition to being a health care professional.

## Supervision and absenteeism

Most health staff had their supervisors present at their workplace, though this was higher in urban areas (99%) than rural areas (90%), and overall satisfaction with supervision was high (75% of the sampled doctors and 88% of the sampled ATS agreed with the statement that their supervisors consider their needs). Most supervisors' initial reaction to their staffs' absenteeism was understanding the reasons for their absence (63–78% of staff across the cadres expect this behaviour), though sanctions and warnings of various kinds were also likely. Only 1–3% of staff across cadres believed that no action would be taken in the event of staff absence.

Staff reported that colleagues were absent for two to three days per month on average, with sickness being the main reason, followed by family-related reasons. Reported absence data suggested about 11% of working days are lost each month, with higher levels for doctors than other staff and higher levels for urban areas (though the difference was not significant).

## Motivational profiles

Health staff showed low levels of amotivation, moderate levels of intrinsic motivation, and high identified and integrated regulation (2.5 points out of 4, which is much higher than the scores for external and introjected regulation). This suggests that the average health worker is motivated most by helping others, having responsibilities, having a sense of meaning in their work, and feeling personally responsible. There were no significant differences across groups in relation to this. Intrinsic motivation, somewhat contrary to expectations, was significantly higher for urban than for rural health staff.

## DCE

For the DCE, the responses were well balanced on the whole, though a small but significant proportion of health staff followed dominant attributes: 10% of doctors always opted for specialisation, while 8.5% always followed the largest wage. For nurses and midwives, some locations dominated: 13% of nurses always picked Conakry as a choice for posting, while 7% always picked a regional hospital; the equivalent for midwives was 22% and 4%, respectively.

We present the DCE results in Table 6. We report the marginal rate of substitution (MRS) between the monetary attribute (in USD per month) and the levels of the non-monetary attributes. This can be interpreted as the willingness to pay, i.e. the monthly wage gain/loss the health worker is willing to take in order to modify the other aspects of their working conditions.

**Table 6. MRS: Monetary value in USD.**

| | Doctors | | Nurses/midwives | | ATS | |
|---|---|---|---|---|---|---|
| | MRS | p-value | MRS | p-value | MRS | p-value |
| **Location (ref. = Conakry)** | | | | | | |
| Regional hospital | 14.64 | 0.725 | 141.56 | 0.079 | 307.30 | 0.004*** |
| Prefectural hospital | -568.45 | <0.001*** | 36.17 | 0.665 | 260.37 | 0.007*** |
| Rural health center | | | -434.88 | 0.001*** | 204.68 | 0.042** |
| Rural health post | | | | | 2.22 | 0.979 |
| **Working condition/equipment (ref. = poor)** | | | | | | |
| Medium | 127.14 | 0.011** | 248.28 | <0.001*** | 22.74 | 0.516 |
| Good | 165.74 | <0.001*** | 296.54 | <0.001*** | 57.47 | 0.107 |
| **Training (ref. = none)** | | | | | | |
| Workshop | 120.59 | 0.002*** | 640.78 | <0.001*** | | |
| Specialisation | 439.21 | <0.001*** | 718.33 | <0.001 | | |
| Nursing school | | | | | 458.07 | <0.001*** |
| **Time-bound (ref. = none)** | | | | | | |
| 5 years | 21.52 | 0.581 | | | | |
| 7 years | | | 62.13 | 0.174 | 5.33 | 0.871 |
| **Housing** | | | | | | |
| Yes | 102.84 | 0.001*** | 126.36 | 0.003*** | 47.15 | 0.264 |
| **Motor-bike** | | | | | | |
| Yes | 68.89 | 0.032** | 215.11 | <0.001*** | 136.90 | <0.001*** |

\* p<0.1,

\*\* p<0.05,

\*\*\* p<0.01.

The DCE analyses indicated that for all health workers, except ATS, location and training were the most significant items (Table 6). For doctors, Conakry and regional hospitals carried about the same value, but the negative cost associated with a prefectural (district) hospital was the largest, estimated at around 568 USD. For nurses and midwives, the lowest-level facility was the rural health centre, for which they would require 435 USD. However, prefectural locations were not significantly different from Conakry or regional hospitals. For nurses and midwives, increasing their human capital was more important: a nursing specialisation and frequent workshops were valued at about 640 and 718 USD per month, which is extremely high. ATS also valued upgrading their skills and status to that of nurses, at an additional 458 USD, which was also the single largest attribute. Specialisation comes second for doctors, at about 439 USD, while training workshops would be worth 121 USD.

In third overall position were the material conditions at the health facility, valued by medical doctors at 127 USD for medium-level and 166 USD for good conditions. Nurses and midwives valued working conditions more, at 248 USD and 297 USD per month on average respectively. Material conditions were not significant for ATS. Good conditions were more appreciated than medium-level conditions by all cadres, but the difference was not statistically significant.

In fourth position was housing, at about 103 USD for doctors, 126 for nurses and midwives, and 47 USD for ATS. The value placed on motorbikes was more varied: doctors value them less than housing, at only 69 USD, while nurses/midwives and ATS value them more than housing, at 215 USD and 137 USD, respectively, per month. The difference between the housing and transportation mean was, however, not statistically significant for any group.

In disaggregated analysis, there was little significance in the various groups of doctors tested. Gender and rural origins do not shape preferences significantly. There were only two attributes which were subject to change: motorbikes and specialisation were preferred by younger doctors (under 44 years) and disliked by older ones. Specialisation was also not sought as much by doctors who were married and have a house, but was aimed at by doctors from a self-reported lower economic status.

There was a lot more diversity in preferences among the nurses and midwives group. In terms of location, those that were married, from rural origins, poorer, older, and male preferred a regional hospital or a prefectural hospital, rather than a location in Conakry. Nurses/midwives from a rural area, of self-assessed poor socio-economic status, and male, even positively valued a posting in a rural health centre. The high value of equipment and specialisation came from the younger cohort and the middle class. A limited-term posting mainly interested the young cohort.

For ATS, there was less heterogeneity in the preferences of different groups. As for other health workers, younger people valued studying more. Socio-economically wealthy people had a strong preference for staying in Conakry. ATS from rural origins valued a good wage, while females valued it less.

**Health workers' natural allocation and scenarios.**   Based on the DCE results and knowledge/assumption about the current levels of the attributes, it is possible to predict the "natural" allocation of health workers in the different location. This is call this a bundle analysis. We first designed a baseline scenario, which consisted of a typical job at each facility level. Applying these assumptions, the natural equilibrium would leave the lower-level facilities understaffed, with only 9% of doctors, 11% of nurses/midwives, and 15% of ATS choosing to work there. This is even lower than the actual allocation. Figs 6 and 7 show the natural allocation of doctors and nurses/midwives in relation to the rural salary (assuming public urban salary is constant). In black bold are displayed the baseline scenarios, i.e. what is closest to the situation at the time of the survey. The other lines show the shift of the curve with a hypothetical rural job improvement.

We proposed a medium-level equipment and workshops for rural jobs scenario and estimate that the share of doctors in prefectural hospitals would increase to 15% (Fig 6). A second improvement would be to give to those who go to rural areas, instead of the workshops, a higher probability of specialisation. The simulation for this gave a very large increase in uptake: 26% (equal to increasing wages by 444 USD). The third upgrade would be a 'brick and mortar' refurbishing of the prefectural-level facilities, as well as building a house for the doctors. This would increase uptake by an additional 6% (~one-third uptake) and would also improve patients' welfare through improved facilities and longer operating hours for emergencies. Finally, the fourth option would be to roughly double the wages of doctors, on top of the housing, good materials, and specialisation possibilities, which was expected to lead to 44% of doctors practising at the prefectural level.

For nurses and midwives, the 'natural' allocation to rural health centres (under the same assumptions as for doctors) would be only 11%, and more than two-thirds would be in Conakry or in a regional hospital. This is not in line with the optimal allocation of health workers. We estimate that providing either workshops or offering specialisation to upgrade their skills, associated with improved equipment, would increase the uptake rate of rural jobs almost threefold (Fig 7). This alone would provide a good balance of the nurses and midwives across different facility types.

The ATS distribution among all levels is not as imbalanced as the market of doctors, nurses, and midwives: at baseline, 21% would work at the rural health centre level, and 15% in health posts. As for other health workers, the skills upgrade (to nurse) would be an

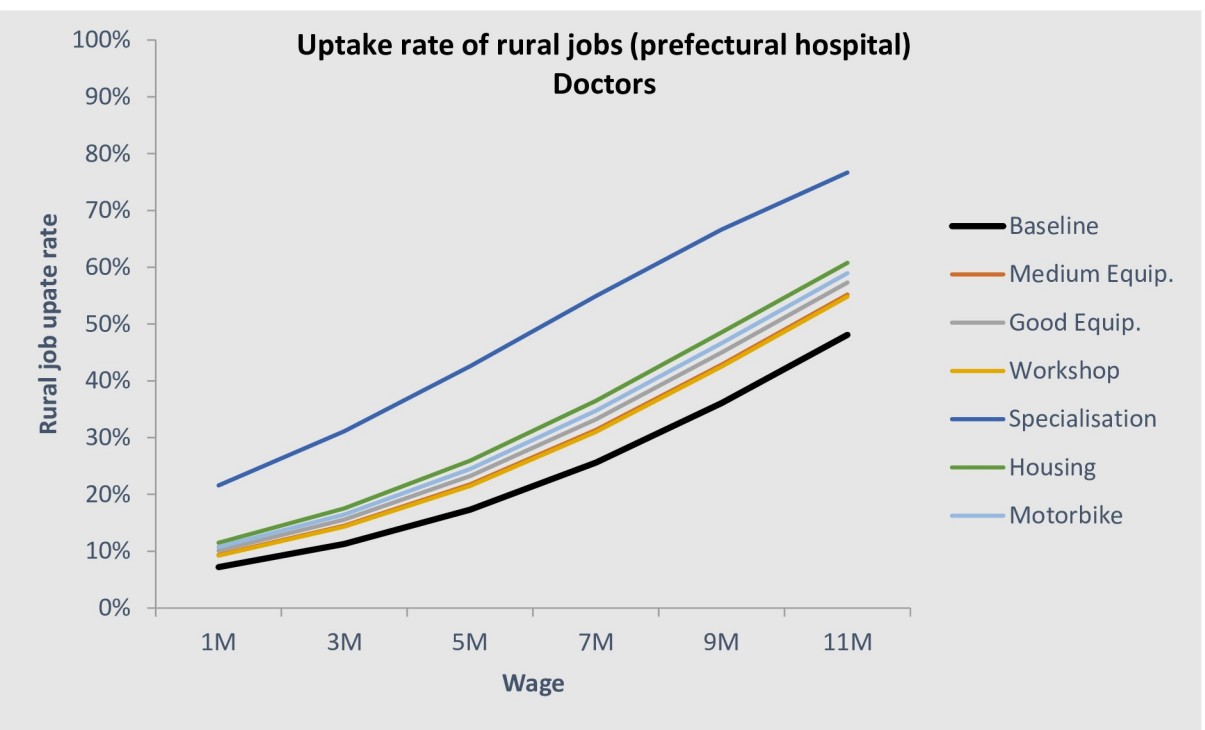

**Fig 6. Uptake rate of prefectural posts for doctors.**

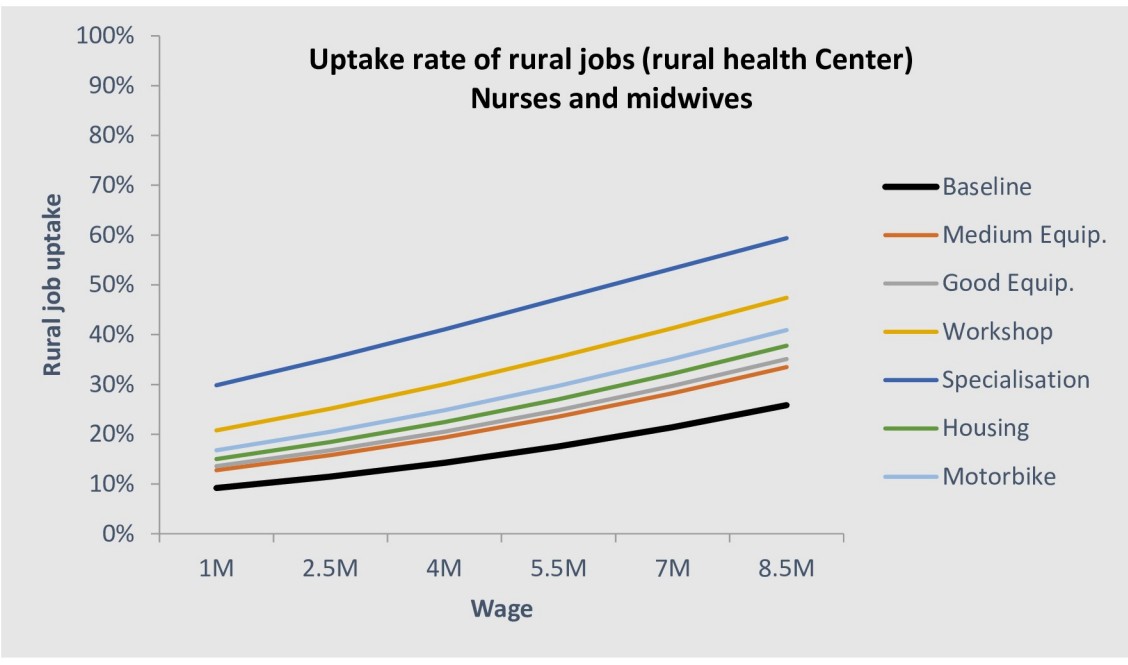

**Fig 7. Uptake rate of rural health centre jobs for nurses and midwives.**

important motivator, bringing uptake rates to 27% in health posts (Fig 8). While equipment or housing would not have a large effect, a wage increase of US$222 would increase uptake by an extra 7%.

## Discussion

This study adopted a complex methodology, combining nationally representative health worker survey, DCE, patient exit interviews and facility surveys, combined to better understand the drivers of health staff motivation, retention, satisfaction and performance in Guinea. In this study we have focussed on the health worker survey and DCE findings. They are broadly compatible in terms of their findings but were designed to be complimentary, with the health worker survey providing understanding of staff's trajectories and current situation, and the DCE focusing on the likely impact of future policy interventions.

Consistent with earlier data from Guinea [6], our findings highlight the high needs of the population and the maldistribution of health staff. The findings highlight some encouraging areas—for example, the relatively high levels of satisfaction of staff and users in many domains —as well as more worrying features, such as the high proportion of staff favouring emigration, and their high tolerance of user payments for services and quality. Areas of concern which are flagged up include the lack of confidence exhibited by a substantial minority of staff in relation to core tasks which they regularly have to carry out, which reinforces existing published evidence on quality of essential health care in Guinea [16], as well as their limited exposure to rural areas during training. Most staff are generally satisfied with their work and with supervision. However, financial aspects and working conditions are considered by them to be least satisfactory. Moreover, from the patient side, we learn that privacy is not well respected in health facilities. Consistent with existing evidence, the high prevalence of out-of-pocket payments by patients is highlighted. Given the low levels of receipting, it is not clear how far these are formal or informal. Some households are driven to risky strategies to pay for healthcare, such as asset sales.

It is relevant to compare the survey findings with qualitative results from our focus group discussions [15]. These highlighted the need to present rural service as a temporary state— many doctors, for example, are unwilling to be posted to rural areas (which for them means any place more than 10 km from a city, and includes prefectural capitals, due to their small

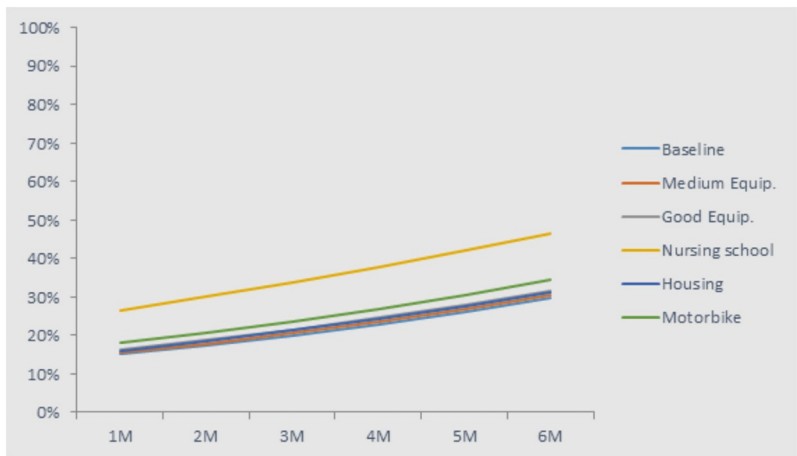

**Fig 8. Uptake rate of jobs in rural health posts for ATS.**

population and limited facilities), as they are seen as places from which it is very hard to leave. Their concerns include professional isolation, poor living and working conditions, and also the difficulty of getting promoted when working in rural areas. The costs of living and housing are higher in these areas, the rural allowance is inadequate to cover this gap, and work opportunities for them and their families are limited. Workloads are regarded as higher in rural areas (as supported by our survey) and training opportunities fewer. Health staff have no health insurance and transportation is challenging. All of these aspects—which are shared in studies addressing this question in other low income and fragile contexts [8, 13, 17–20]—need to be addressed in order to retain doctors at the prefectural level.

Certain policy responses emerge as promising from the surveys, DCE and the qualitative data collection: in particular, offering upgrading and specialisation in return for rural service; providing greater exposure to rural areas during training; increasing recruitment from rural areas; experimenting with fixed-term contracts in rural areas, especially for younger staff; improving working conditions in rural posts; and attracting staff with high intrinsic or external introjected motivation.

In terms of recruiting and retaining health staff in rural areas: for doctors, it is clear from the DCE that conditional specialisation options, particularly for younger doctors, could be a powerful motivator to attract and retain doctors at prefectural hospital level, followed by investment in facility equipment and provision of housing and transport. This is consistent with findings in other contexts about the importance of specialisation for doctors [21]. This may require regulatory adaptation to civil service rules in Guinea.

To motivate nurses and midwives to serve at the health centre level would require, according to our findings, training options linked to rural services, alongside investment in moderate levels of equipment, transport, and housing. These actions are more likely to be effective for staff from poorer backgrounds and rural areas.

For ATS, offering upgrading training after a period of service in rural areas, especially for younger staff, could be an effective motivator, alongside the offer of transport and housing. Staff from poorer backgrounds, and potentially women, may be more attracted by this package.

The study has some limitations. We did not attempt to survey community health workers. Although the community health workers do perform important functions in Guinea, the study focused on the formal health workforce. We did not attempt to survey staff working purely in the private sector, as this was not our focus, although our tools do shed important light on health staff dual practice and attitudes towards working in the private versus public sector. The focus here was on generating largely quantitative analysis on staff preferences and policy-modifiable factors influencing their choices on location in particular. For these purposes, our tools were adequate, though in future additional qualitative interviews at local and national level could help shape our understanding of what margins there are for human resource package reforms.

This study has not examined cost-effectiveness, which should be further considered as the Ministry of Health develops its national human resources strategy. It is likely that a focus on supporting mid-level cadres in rural areas will be a priority, given the limited funding and existing extreme maldistribution. International evidence suggests that well-supported mid-level cadres (such as ATS) can provide appropriate healthcare in rural areas [10]. Addressing the health workforce imbalance, in Guinea as in other stressed settings, also requires a good understanding of what is feasible within the local political economic context [22], which will require further elaboration in the context of plans in Guinea to allocate funds from mining industries to local authorities.

The study has also not considered the issue of how posts and funds are allocated from the MoH side, as our objective was to understand health workers' perceptions, motivation, and

preferences. The wider evidence base highlights the importance of ensuring funding is made available for rural health posts and is managed in a decentralised way, so that health staff who do not take up positions in these areas cannot continue on the public payroll in urban areas [10]. Local performance management is likely to be more effective and supportive in other ways too, and is consistent with the wider regional and international emphasis on decentralisation, accountability, and resource allocation according to results [23]. This will require changes to public funding, as well as capacity building and improved incentives for local-level managers in the public sector. The political economy of reforms in this area are complex as other studies have highlighted, with the Ministry of Finance and Ministry of Public Services controlling payment and recruitment of health staff respectively and external actors also playing an important role in shaping human resources policies (24). On the other hand, crises such as Ebola—and presumably now COVID-19 –offer a potential window of opportunity to address long-standing distribution and staff mix challenges [11, 24].

## Conclusions

The objective of this study was to provide analysis to inform more effective human resource policy making and improve the distribution of staff to rural areas, in particular. This policy challenge is common to many countries, but especially acute in low income and fragile states, where the levers to manage the public health care market effectively are typically weak. The research also aims to add to the hitherto limited research on human resources for health in francophone Africa.

Our findings highlight some encouraging findings around staff satisfaction, including with supervision and non-financial aspects of their work. However, they also raise concerns around staff preparedness for their roles, future career intentions and informal charging. Options for better rural attraction and retention include offering upgrading and specialisation in return for rural service (especially for doctors); providing greater exposure to rural areas during training; increasing recruitment from rural areas; experimenting with fixed-term contracts in rural areas, especially for younger staff; improving working conditions in rural posts; and attracting staff with high intrinsic or external introjected motivation. However, the development of incentive packages should be accompanied by action to tackle the wider issues too, through reforms to training and decentralisation of human resources funding and management, considering all stages of the human resource journey and with sensitivity to the different staff profiles and preferences expressed in this study.

## Supporting information

**S1 File. Health staff survey English.**
(PDF)

**S2 File. Health staff survey French.**
(PDF)

**S3 File. Summary of user perspectives and facility survey.**
(DOCX)

**S4 File. Health worker dataset.**
(DTA)

## Acknowledgments

We would like to thank all participants and the enumerators of this study. We would also particularly like to acknowledge the contribution of Professor Juan Munoz on sample design and sampling, Dr Yushuf Sharker for the guidance on statistical analysis and interpretation, Hanna Laufer, Zezhen Wu and Brian Law for data capture and analysis, and of Dr Mohammed Faza and Dr Yeroboye Camara of the Ministry of Health, Guinea to study design and overarching guidance.

## Author Contributions

**Conceptualization:** Sophie Witter, Christopher H. Herbst, Marc Smitz, Ibrahim Magazi, Rashid U. Zaman.

**Data curation:** Mamadou Dioulde Balde.

**Formal analysis:** Sophie Witter, Marc Smitz, Rashid U. Zaman.

**Funding acquisition:** Christopher H. Herbst, Ibrahim Magazi.

**Investigation:** Sophie Witter, Mamadou Dioulde Balde.

**Methodology:** Sophie Witter, Christopher H. Herbst, Marc Smitz, Mamadou Dioulde Balde, Rashid U. Zaman.

**Project administration:** Rashid U. Zaman.

**Resources:** Ibrahim Magazi.

**Supervision:** Sophie Witter, Mamadou Dioulde Balde, Rashid U. Zaman.

**Validation:** Mamadou Dioulde Balde.

**Visualization:** Rashid U. Zaman.

**Writing – original draft:** Sophie Witter, Marc Smitz.

**Writing – review & editing:** Sophie Witter, Christopher H. Herbst, Marc Smitz, Mamadou Dioulde Balde, Ibrahim Magazi, Rashid U. Zaman.

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
