## [Decision Letter · Decision Letter 0]

17 May 2021

PONE-D-20-40985

How to attract and retain health workers in rural areas of a fragile state: findings from a labour market survey in Guinea

PLOS ONE

Dear Dr. Witter,

Thank you for submitting your manuscript to PLOS ONE. After careful consideration, we feel that it has merit but does not fully meet PLOS ONE’s publication criteria as it currently stands. Therefore, we invite you to submit a revised version of the manuscript that addresses the points raised during the review process.

We look forward to receiving your revised manuscript.

Kind regards,

Ali B. Mahmoud, Ph.D.

Academic Editor

PLOS ONE

**Additional Editor Comments (if provided):**

Please ensure the novelty of this work is visible more substantially in your revised manuscript, mainly by acknowledging the contemporary work in your research area.

Journal Requirements:

2)  Please provide additional details regarding participant consent. In the ethics statement in the Methods and online submission information, please ensure that you have specified what type you obtained (for instance, written or verbal, and if verbal, how it was documented and witnessed). If your study included minors, state whether you obtained consent from parents or guardians. If the need for consent was waived by the ethics committee, please include this information.

For additional information about PLOS ONE ethical requirements for human subjects research, please refer to " ext-link-type="uri" xlink:type="simple">http://journals.plos.org/plosone/s/submission-guidelines#loc-human-subjects-research."

3) We note that you have reported significance probabilities of 0 in places. Since p=0 is not strictly possible, please correct this to a more appropriate limit, eg 'p0.0001'.

4) Please include additional information regarding the survey or questionnaire used in the study and ensure that you have provided sufficient details that others could replicate the analyses. For instance, if you developed a questionnaire as part of this study and it is not under a copyright more restrictive than CC-BY, please include a copy, in both the original language and English, as Supporting Information.

5) Please provide some clarification whether the ethics approval was obtained for the preliminary qualitative work conducted.

6) Please ensure that you refer to Figures 7 8 in your text as, if accepted, production will need this reference to link the reader to the figures.

7) Thank you for stating the following in the Competing Interests section:

[The authors have declared that no competing interests exist.].   

We note that one or more of the authors are employed by a commercial company: Oxford Policy Management

i. Please provide an amended Funding Statement declaring this commercial affiliation, as well as a statement regarding the Role of Funders in your study. If the funding organization did not play a role in the study design, data collection and analysis, decision to publish, or preparation of the manuscript and only provided financial support in the form of authors' salaries and/or research materials, please review your statements relating to the author contributions, and ensure you have specifically and accurately indicated the role(s) that these authors had in your study. You can update author roles in the Author Contributions section of the online submission form.

ii. Please also provide an updated Competing Interests Statement declaring this commercial affiliation along with any other relevant declarations relating to employment, consultancy, patents, products in development, or marketed products, etc.  

Reviewers' comments:

Reviewer's Responses to Questions

**Comments to the Author**

1. Is the manuscript technically sound, and do the data support the conclusions?

Reviewer #1: Yes

Reviewer #2: Yes

2. Has the statistical analysis been performed appropriately and rigorously? 

Reviewer #1: I Don't Know

Reviewer #2: Yes

3. Have the authors made all data underlying the findings in their manuscript fully available?

Reviewer #1: Yes

Reviewer #2: Yes

4. Is the manuscript presented in an intelligible fashion and written in standard English?

Reviewer #1: Yes

Reviewer #2: Yes

5. Review Comments to the Author

Reviewer #1: Summary of study

This study uses three different nationally representative cross-sectional surveys as well as a discrete choice experiment to examine health workers’ motivations and how they could be incentivized to work in remote rural areas of Guinea. The authors examine differences by cadre of health worker as well as by location They suggest several ways to improve health worker retention in rural areas including offering specializations, recruitment from rural areas and improving working conditions.

Major Comments

1. While I appreciate the authors’ efforts to be thorough and to use different sources of data to examine their research question, I found the paper a little overwhelming. With results from three different surveys, a DCE, a focus on multiple outcomes disaggregated by different cadres and location, it was very difficult to determine what the main findings were of this paper. I suggest the authors simplify and split this up into multiple papers. For example, perhaps they could just focus either on the health worker survey or on the DCE in this paper. At a minimum, I think they should exclude the patient survey and health facility survey as I don’t believe they add substantially to the question of health worker motivation.

2. I think the Methods Section could use a little more detail. For example, how were the questions that are presented in Figure 2-5 asked? Were the respondents presented with responses which they had choose from? What kind of regression models were run for the analyses in Table 1 3 and for the DCE? The results section mentions a composite satisfaction index- how was this created? Similarly, what do the measures of intrinsic/extrinsic motivation consist of? (The authors refer to another paper, but I think a summary here would be useful).

3. It would be useful to have a Table 1 that shows some descriptive statistics for the health workers surveyed; some of these statistics are cited in the text but I could not find this information in the tables.

4. Table 4: It would be helpful if the authors could say a little more about how to interpret the coefficients in this table.

5. The figures could use more labeling so that one can see what they are showing without referring to the text. For example, in Figure 2-5, it’s not clear what the question was. Figures 6 and 7 have no axis titles and, again, it’s not clear what they are showing.

6. If the authors choose to keep both the health worker survey and the DCE in the same paper I think it would be useful to have more discussion of to what extent the results of the health worker survey and the DCE correspond with each other.

Minor Comments

7. Table 1 and 3 are a little overwhelming: it may be helpful to have some kind of headings/categories to help orient the reader.

8. Box 1: Suggest making this into a table so can see all the attributes and levels together.

9. The description of the sampling procedure is quite complicated- perhaps a figure will help the reader?

Reviewer #2: Dear Authors,

Thank you for this paper, conducting a Labor market survey in Guinea. Actually i had already seen an earlier version and presentation of it, shared by Prof. Witter.

My main comments are the following; This is a proper conducted health labour market analysis of the health sector , its incentives , constraints and regulation in Guinea. As a method, amongst others, A DCE has been used. Whilest this has been conducted properly, i miss consinderable 'contextual' information and 'the policy momentum' post-ebola that enabled, but also distored , workforce developments and activities in Guinea. context and health system are 'fragile' in a sense that domestic political, epidemic (Ebola, covid-19, but also new measles and polio outbreaks) as well as international actor involvement have 'shaped' the labour market in several ways. This specific Guinean governance context and momentum (post-ebola workforce investments) should be explained more explicit in introduction and discussion as it impacts specificity, relevance and policy uptake of this analysis.

As 'fragility' is in the title of the article, it must be more explicit explained what this fragility implies for the health and labour sector in Guinea

We wrote partly about it here but several more articles should be available

Kolie, D., Delamou, A., van de Pas, R., Dioubate, N., Bouedouno, P., Beavogui, A. H., ... Van Damme, W. (2019). ‘Never let a crisis go to waste’: post-Ebola agenda-setting for health system strengthening in Guinea. BMJ global health, 4(6), e001925.

Kolie D, van de Pas R, Delamou A, Dioubaté N, Beavogui FT, Kaba A, Beavogui AH, MD, Van De Put W, Van Damme W. Retention of Healthcare Workers one year after recruitment and deployment in rural settings: an experience Post-Ebola in five Health Districts in Guinea. Hum Resour Health. 2021

https://doi.org/10.21203/rs.3.rs-122033/v1

Bests and looking forward to a next version. Remco van de Pas

6. PLOS authors have the option to publish the peer review history of their article (what does this mean?). If published, this will include your full peer review and any attached files.

Reviewer #1: No

Reviewer #2: **Yes: **Remco van de Pas

---

## [Decision Letter · Decision Letter 1]

16 Aug 2021

PONE-D-20-40985R1

How to attract and retain health workers in rural areas of a fragile state: findings from a labour market survey in Guinea

PLOS ONE

Dear Dr. Witter,

Thank you for submitting your manuscript to PLOS ONE. After careful consideration, we feel that it has merit but does not fully meet PLOS ONE’s publication criteria as it currently stands. Therefore, we invite you to submit a revised version of the manuscript that addresses the points raised during the review process.

The authors have done a good job in addressing the comments of the reviewers, yet there remain a number of points that would need to be addressed prior to consideration for publication.

If applicable, we recommend that you deposit your laboratory protocols in protocols.io to enhance the reproducibility of your results. Protocols.io assigns your protocol its own identifier (DOI) so that it can be cited independently in the future. For instructions see: http://journals.plos.org/plosone/s/submission-guidelines#loc-laboratory-protocols. Additionally, PLOS ONE offers an option for publishing peer-reviewed Lab Protocol articles, which describe protocols hosted on protocols.io. Read more information on sharing protocols at https://plos.org/protocols?utm_medium=editorial-emailutm_source=authorlettersutm_campaign=protocols.

We look forward to receiving your revised manuscript.

Kind regards,

Mohamad Alameddine, MPH, Ph.D.

Academic Editor

PLOS ONE

Journal Requirements:

Reviewers' comments:

Reviewer's Responses to Questions

**Comments to the Author**

1. If the authors have adequately addressed your comments raised in a previous round of review and you feel that this manuscript is now acceptable for publication, you may indicate that here to bypass the “Comments to the Author” section, enter your conflict of interest statement in the “Confidential to Editor” section, and submit your "Accept" recommendation.

Reviewer #1: (No Response)

Reviewer #2: All comments have been addressed

2. Is the manuscript technically sound, and do the data support the conclusions?

Reviewer #1: Yes

Reviewer #2: Yes

3. Has the statistical analysis been performed appropriately and rigorously? 

Reviewer #1: Yes

Reviewer #2: (No Response)

4. Have the authors made all data underlying the findings in their manuscript fully available?

Reviewer #1: Yes

Reviewer #2: Yes

5. Is the manuscript presented in an intelligible fashion and written in standard English?

Reviewer #1: Yes

Reviewer #2: (No Response)

6. Review Comments to the Author

Reviewer #1: I appreciate the authors efforts to respond to my comments, I think the paper is improved and easier to follow. A few additional comments/suggestions below:

1. Table 1: For the time-bound contract attribute, could the authors clarify what the different levels were? Were the levels increments of a year up to the maximum of 5/7 years? Or something else?

2. I don’t find Table 2 particularly useful, given that it just shows the geographical distribution of the health workers. Instead (or in addition), I think it would be useful to show the descriptive statistics that are mentioned throughout the paper but aren’t shown anywhere. For example, I think the table could show summary statistics cited in the text on work/career history, workload (hours per week, patients seen), satisfaction scores, remuneration etc by health worker type. I think this would help the reader to be able to quickly see differences between the different health worker types.

3. Are the regressions in Table 3 and Table linear models? If so, it would be good to mention this in the methods section.

4. Table 3: For the satisfaction composite score- is it actually a dissatisfaction score? (it seems like a negative coefficient actually means higher satisfaction, correct?). If so, I think it would be helpful to label it as such.

5. If Table 5 presents the results of a linear model, I think the coefficients should be interpreted in terms of percentage points and not percentages.

6. In Lines 321-322, the authors say that those with higher intrinsic motivation are more likely to want to work in rural areas in the long-term, however it should be noted that this isn’t statistically significant.

Reviewer #2: Amendments properly include suggestions from reviewer 1 and 2. Would encourage authors to be engaged in followup research and processed regarding workforce development in Guinea given needs and development. This labor market analysis is an important basis for continuing work.

7. PLOS authors have the option to publish the peer review history of their article (what does this mean?). If published, this will include your full peer review and any attached files.

Reviewer #1: No

Reviewer #2: **Yes: **Remco van de Pas

---

## [Author Response · Author response to Decision Letter 1]

10 Sep 2021

We have tried to respond fully to these additional comments - please see response letter for details.

---

## [Decision Letter · Decision Letter 2]

5 Nov 2021

How to attract and retain health workers in rural areas of a fragile state: findings from a labour market survey in Guinea

PONE-D-20-40985R2

Dear Dr. Witter,

We’re pleased to inform you that your manuscript has been judged scientifically suitable for publication and will be formally accepted for publication once it meets all outstanding technical requirements.

Kind regards,

Elizabeth S. Mayne, M.D.

Academic Editor

PLOS ONE

Additional Editor Comments (optional):

Reviewers' comments:

Reviewer's Responses to Questions

**Comments to the Author**

1. If the authors have adequately addressed your comments raised in a previous round of review and you feel that this manuscript is now acceptable for publication, you may indicate that here to bypass the “Comments to the Author” section, enter your conflict of interest statement in the “Confidential to Editor” section, and submit your "Accept" recommendation.

Reviewer #1: (No Response)

Reviewer #2: All comments have been addressed

2. Is the manuscript technically sound, and do the data support the conclusions?

Reviewer #1: Yes

Reviewer #2: Yes

3. Has the statistical analysis been performed appropriately and rigorously? 

Reviewer #1: Yes

Reviewer #2: N/A

4. Have the authors made all data underlying the findings in their manuscript fully available?

Reviewer #1: Yes

Reviewer #2: (No Response)

5. Is the manuscript presented in an intelligible fashion and written in standard English?

Reviewer #1: Yes

Reviewer #2: Yes

6. Review Comments to the Author

Reviewer #1: Thank you for the changes, I appreciate all the edits and efforts to respond to the comments. My only remaining concern is that I still believe that the coefficients in Table 5 should be reported as percentage points and not percentages if the regression model was a linear regression and the outcome is binary. However, I may have mis-understood the regression model used.

Reviewer #2: (No Response)

7. PLOS authors have the option to publish the peer review history of their article (what does this mean?). If published, this will include your full peer review and any attached files.

Reviewer #1: No

Reviewer #2: **Yes: **Remco van de Pas

---

## [Editor Report · Acceptance letter]

7 Dec 2021

PONE-D-20-40985R2 

How to attract and retain health workers in rural areas of a fragile state: findings from a labour market survey in Guinea 

Dear Dr. Witter:

I'm pleased to inform you that your manuscript has been deemed suitable for publication in PLOS ONE. Congratulations! Your manuscript is now with our production department. 

Kind regards, 

on behalf of

Dr. Elizabeth S. Mayne 

Academic Editor

PLOS ONE